# Individual-based model highlights the importance of trade-offs for virus-host population dynamics and long-term co-existence

**Fateme Pourhasanzade**[1], **Swami Iyer**[2], **Jesslyn Tjendra**[1], **Lotta Landor**[1], **Selina Våge**[1] *

**1** Department of Biological Sciences, University of Bergen, Bergen, Norway, **2** Computer Science Department, University of Massachusetts, Boston, Massachusetts, United States of America

* selina.vage@uib.no

**Data Availability Statement:** All relevant data are within the manuscript and its Supporting information files. An implementation of the IBM in

## Abstract

Viruses play diverse and important roles in ecosystems. In recent years, trade-offs between host and virus traits have gained increasing attention in viral ecology and evolution. However, microbial organism traits, and viral population parameters in particular, are challenging to monitor. Mathematical and individual-based models are useful tools for predicting virus-host dynamics. We have developed an individual-based evolutionary model to study ecological interactions and evolution between bacteria and viruses, with emphasis on the impacts of trade-offs between competitive and defensive host traits on bacteria-phage population dynamics and trait diversification. Host dynamics are validated with lab results for different initial virus to host ratios (VHR). We show that trade-off based, as opposed to random bacteria-virus interactions, result in biologically plausible evolutionary outcomes, thus highlighting the importance of trade-offs in shaping biodiversity. The effects of nutrient concentration and other environmental and organismal parameters on the virus-host dynamics are also investigated. Despite its simplicity, our model serves as a powerful tool to study bacteria-phage interactions and mechanisms for evolutionary diversification under various environmental conditions.

## Author summary

Genetic diversification in microbial communities is an important process with far-reaching consequences both for ecosystem functioning and public health. Yet, the mechanisms governing the selection of new microbial strains in ecosystems as well as developing infectious diseases are still relatively poorly understood. The sheer diversity in both bacterial and viral communities begs for a conceptual understanding of these regulatory mechanisms. Here we study one such presumably important mechanism, namely the trade-off between the host's growth and thus competitive abilities and its abilities to defend against the virus. To that end, we introduce an idealized individual-based model of bacterial and

the Python programming language is available at a public Github repository [24].

**Funding:** The work was funded by Trond Mohn Research Foundation (https://mohnfoundation.no/?lang=en) stipend TMS2018REK02 awarded to SV. The funders had no role in study design, data collection and analysis, decision to publish, or preparation of the manuscript.

**Competing interests:** The authors have declared that no competing interests exist.

viral community to study the effects of trade-off based versus random interactions on short- and long-term population dynamics. Short-term infection dynamics emerging from our model are validated with experimental data. Our simulations show that long-term co-existence of the virus and host critically depends on the nature of trade-off regulating the virus-host interactions. Specifically, highest diversity in both host and viral communities and co-existence over long time scales are favored in regimes of trade-off based compatibility between viruses and their hosts.

## Introduction

Viruses execute a wide range of functions in the biosphere, influencing biogeochemical cycles, affecting efficiencies of transport of energy and matter through food-webs and driving processes of evolutionary diversification [1–4]. These ecosystem-related functions may seem disentangled from the effects that pathogenic viruses can have on human health and society as demonstrated by the ongoing COVID-19 pandemic [5]. Yet, the underlying mechanisms driving the dynamics between viruses and their hosts are in principle the same.

Viral ecology has been growing as a field of research in the last decades, with sequencing techniques revealing enormous biodiversity in viral genomes [6]. The field, however, remains challenging, since advanced and at times intricate laboratory experiments are required to characterize viral traits as well as interactions with their hosts. Besides, much of the environmental viral metagenome remains unmapped or still undiscovered [7]. We conjecture that virology will grow as a field if we manage to focus on principles that unify different disciplines, from viral ecology to infectious disease research. To better understand the fundamental mechanisms that drive viral dynamics, conceptual models should be used as tools to identify principles that explain biodiversity and functioning in viral systems, be they ecological feedback mechanisms or emerging evolutionary dynamics.

Trait-based approaches have proved useful in identifying unifying and universally applicable mechanisms in ecology and were first established in terrestrial ecology [8, 9]. They have also been successfully applied to marine ecology [10], and in particular to microbial ecology [11–13]. A strength of trait-based approaches is that processes and interactions between organisms and their environment are described on a functional level, independent of particular taxa at hand. Besides, it brings trade-offs between organismal traits to the center of attention, which arise from chemical and physical constraints that are universal (ie, system-independent). Such trade-offs are crucial to understanding life [14–18]. In the marine microbial ecosystem, various processes and structures have been linked through fundamental trade-offs [19]. Trait- and trade-off based perspectives thus also promise to unify viral ecology [20].

In the present context, we use a simple evolutionary individual-based model to test the hypotheses that trade-offs between competitive and defensive traits are key to understanding virus-host population dynamics and evolutionary change. We define competitive traits as organism-specific traits to acquire limiting resources (in our case, nutrient affinities of hosts) and defensive traits as traits modifying the efficiency of a viral infection (here, the viral adsorption coefficient and the inverse of the host range of viruses). Following [21], we model each of these traits as genes being embedded in a phenotypic trait function, and we incorporate trade-offs between them by means of interaction functions that determine the likelihood of infection based on gene similarity of the host and virus ("compatibility function") and virus-intrinsic infection efficiency ("virulence function"). The model is highly idealized, with emphasis on

infection mechanisms that are important and applicable to both environmental as well as pandemic settings.

In the next section, we describe our individual-based model along with experimental methods used to characterize virus-host infection dynamics. This is followed by findings from our model, including a validation of the model dynamics with experimental results. We conclude the article by discussing the relevance and generality of our results for increased insights into virus-host interactions and evolutionary dynamics.

## Materials and methods

### Individual-based model

Individual-based models (IBMs) are in-silico models that describe the behavior of autonomous individuals (organisms). These models are widely used, not only in ecology [22] but also in other disciplines dealing with complex systems made up of autonomous entities [23]. In this section, we give an informal description of an evolutionary IBM to study the interaction patterns between bacteria and virus. For a formal description of the algorithm used in the IBM, we refer the reader to the pseudocode for the IBM in the Supporting Information section. An implementation of the IBM in the Python programming language is available at a public Github repository [24].

The state variables and parameters (environmental and organismal) of the IBM are listed in Table 1, each with a symbol, description, typical value [21], and units.

For simplicity, a single elemental resource is used in the model budget. Specifically, phosphorous is used as the model currency [25]. For ease of interpretation of the results, the total phosphorus concentration $P$ is expressed in terms of the number of host individuals in the simulated volume, ie, $P \sim P/(P_h \cdot V)$. Similarly, the nutrient affinity $\alpha$ of a host and the adsorption coefficient $\beta$ of a virus are normalized in terms of the simulated volume, ie, $\alpha \sim \alpha/V$ and $\beta \sim \beta/V$.

At the start of the simulation, we consider a nutrient medium of volume $V$. The medium is inoculated with a host population $\mathbf{H} = \{h_1, h_2, \ldots, h_{N_h^0}\}$, in which each host $h_i$ has genotype $\{g_\alpha^0\}$ and mass $m_i$ picked uniformly at random from the interval $\left[\frac{1}{2}, 1\right]$. The medium is also inoculated with a virus population $\mathbf{V} = \{v_1, v_2, \ldots, v_{N_v^0}\}$, in which each virus $v_i$ has genotype $\{g_v^0, g_\beta^0\}$. Note that $\mathbf{H}$ and $\mathbf{V}$ are sets of individuals, which grow (or shrink) as dynamics unfold. The initial amount of dissolved phosphorus $P_d^0$ is calculated in units of host individuals as $P_d^0 = P - \Sigma_{i=1}^{N_h^0} m_i$.

At each time step $t \in [1, T]$ of the simulation, we update the current concentration $P_d$ of dissolved phosphorus as $P_d = P_d + \omega(P_d^0 - P_d)$, which takes into account the inflow and outflow of phosphorus due to washout. We then carry out a round of host dynamics followed by a round of virus-host interaction dynamics as described below. Each time step $t$ is fixed to 1 hour. With time-steps lasting 1 hour, rates in units $h^{-1}$ can be translated into probabilties per time step.

During the host dynamics round, we consider each host $h_i \in \mathbf{H}$ in sequence. The host can be washed out of the medium with probability $\omega$. If it remains in the medium, it can die with probability $\delta_h$, in which case its biomass $m_i$ is returned to the medium, ie, $P_d$ is incremented to $P_d + m_i$. If the host does not die, it experiences a mass loss due to metabolism given by $\epsilon_h m_i$, which is subtracted from $m_i$ and added to $P_d$. The host $h_i$ grows during the time step $t$, and the resulting gain in mass is calculated as $\frac{\alpha g_\alpha P_d}{1 + \frac{\alpha g_\alpha P_d}{\mu_h}}$ [26], which is subtracted from $P_d$ and added to $m_i$.

If $m_i$ exceeds unity, then the host divides into two daughter cells, each with half the mass (ie,

**Table 1. State variables and parameters of the IBM.**

| Symbol | Description | Value | Units |
|---|---|---|---|
| | **State variables** | | |
| $N_h$ | host abundance | variable | individuals |
| $N_v$ | virus abundance | variable | individuals |
| | **Environmental parameters** | | |
| $T$ | simulation duration | $3.6 \times 10^1$ | h |
| $t$ | time step duration | 1 | h |
| $V$ | chemostat volume | $10^{-6}$ | L |
| $P$ | total phosphorus concentration | $7.1 \times 10^1$ | $\mu$mol-P L$^{-1}$ |
| $P_d$ | dissolved phosphorus concentration | variable | $\mu$mol-P L$^{-1}$ |
| $\omega$ | chemostat dilution rate | $2 \times 10^{-1}$ | h$^{-1}$ |
| $N_h^0$ | initial value for $N_h$ | $4.6 \times 10^1$ | individuals |
| $N_v^0$ | initial value for $N_v$ | $8.1 \times 10^2$ | individuals |
| $VHR$ | virus to host ratio (the ratio $N_v^0/N_h^0$) | $1.76 \times 10^1$ | - |
| | **Organismal parameters** | | |
| $P_h$ | maximum phosphorus concentration in a host | $8.3 \times 10^{-8}$ | $\mu$mol-P |
| $\alpha$ | nutrient affinity of a host | $1.6 \times 10^{-7}$ | L h$^{-1}$ |
| $\mu_h$ | maximum growth rate of a host | $7.38 \times 10^{-1}$ | h$^{-1}$ |
| $g_\alpha$ | nutrient affinity gene of a host | variable $\in [0, 1]$ | - |
| $g_\alpha^0$ | initial value for $g_\alpha$ | $10^{-1}$ | - |
| $m$ | mass of a host | variable $\in \left[\frac{1}{2}, 1\right]$ | - |
| $\pi_h$ | probability of mutation of host genotype | $6 \times 10^{-2}$ | - |
| $\sigma_h$ | standard deviation of host genotype mutations | $5 \times 10^{-2}$ | - |
| $\delta_h$ | mortality rate of a host | $1.4 \times 10^{-2}$ | h$^{-1}$ |
| $\epsilon_h$ | metabolic loss rate of a host | $1.4 \times 10^{-2}$ | h$^{-1}$ |
| $\beta$ | adsorption coefficient of a virus | $6.2 \times 10^{-11}$ | L h$^{-1}$ |
| $g_v$ | memory gene of a virus | variable $\in [0, 1]$ | - |
| $g_v^0$ | initial value of $g_v$ | $10^{-1}$ | - |
| $g_\beta$ | adsorption coefficient gene of a virus | variable $\in [0, 1]$ | - |
| $g_\beta^0$ | initial value of $g_\beta$ | $10^{-1}$ | - |
| $\pi_v$ | probability of mutation of virus genotype | $6 \times 10^{-3}$ | - |
| $\sigma_v$ | standard deviation of virus genotype mutations | $5 \times 10^{-3}$ | - |
| $\kappa$ | number of viruses produced per infection of a host | $10^1$ | - |
| $\delta_v$ | decay rate of a virus | $1.4 \times 10^{-2}$ | h$^{-1}$ |

Symbol, description, typical value [21], and units.

$m_i/2$) and the same genotype (ie, $\{g_\alpha\}$) as the host. With probability $\pi_h$, the genotype of the daughter cells mutates to $\{g'_\alpha\}$, where $g'_\alpha$ is sampled from a Gaussian distribution with mean $g_\alpha$ and standard deviation $\sigma_h$; and with probability $1 - \pi_h$, the genotype of the daughter cells remains the same as that of the parent cell. The daughter cells are added to the host population $\mathbf{H}$ and the parent cell is removed from it.

During the virus-host interaction round, we consider each virus-host pair $(v_i, h_j)$, where $v_i \in \mathbf{V}$ and $h_j \in \mathbf{H}$. The virus $v_i$ can be washed out of the medium with probability $\omega$. If it remains in the medium, it can decay with probability $\delta_v$. If the virus does not decay, then whether or not it infects the host $h_j$ depends on its compatibility with the host and its virulence. The former is given by the compatibility function $\mathcal{C}$ and the latter by the virulence function $\mathcal{V}$. The

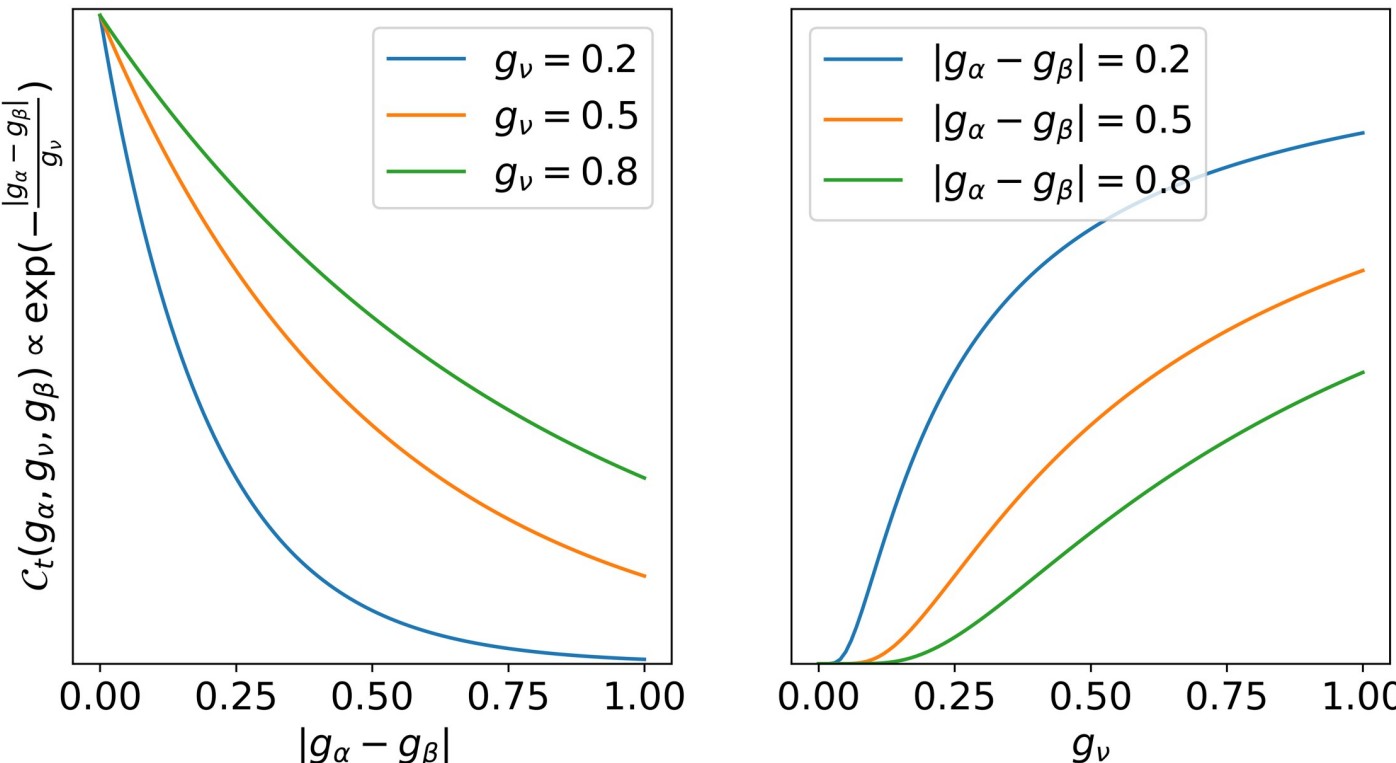

**Fig 1. Trade-off based compatibility function $\mathcal{C}_t$.** The left panel shows how $\mathcal{C}_t$ varies with $|g_\alpha - g_\beta|$ for fixed values of $g_v$; $\mathcal{C}_t$ is high when $g_\alpha$ and $g_\beta$ have similar values and low otherwise. The right panel shows how $\mathcal{C}_t$ varies with $g_v$ for fixed values of $|g_\alpha - g_\beta|$; $\mathcal{C}_t$ is high for high values of $g_v$ and low otherwise.

values of both functions (also called interaction functions) are from the unit interval [0, 1], and are interpreted as probabilities. To test the effects of trade-off based vs random compatibility between hosts and viruses, we consider two compatibility functions: trade-off based compatibility $\mathcal{C}_t(g_\alpha, g_v, g_\beta) = \exp\left(-\frac{|g_\alpha - g_\beta|}{g_v}\right)$ and random compatibility $\mathcal{C}_r$, which samples a number uniformly at random from the unit interval. The function $\mathcal{C}_t$ (Fig 1) captures the trade-off between the host's nutrient affinity and the virus' adsorption coefficient, such that the compatibility is highest when the corresponding gene values $g_\alpha$ and $g_\beta$ are similar. The virus-host compatibility is also enhanced when the virus' host range (expressed by its memory gene $g_v$) is high.

We also consider two virulence functions: trade-off based virulence $\mathcal{V}_t(g_v, g_\beta) = \frac{\beta g_\beta}{g_v}$ and random virulence $\mathcal{V}_r$, which samples a number uniformly at random from the unit interval. The function $\mathcal{V}_t$ (Fig 2) captures the trade-off between the virus' adsorption coefficient $g_\beta$ and its host range $g_v$, whereby virulence of the infection is high for viruses with high adsorption (ie, high $g_\beta$) and narrow host range (ie, small $g_v$). The virus $v_i$ is compatible with host $h_j$ with probability $\mathcal{C}$. If they are compatible, the virus can infect the host with probability $\mathcal{V}$. All viruses are assumed to by lytic, meaning that hosts die upon infection. If the host is infected, then it is removed from the host population **H**, its biomass is immediately recirculated into the dissolved nutrient pool $P_d$, and the virus produces $\kappa$ copies of itself, each of which has the same genotype (ie, $\{g_v, g_\beta\}$) as the parent. With probability $\pi_v$, the genotype of the copies mutates to $\{g_v', g_\beta'\}$, where $g_v'$ and $g_\beta'$ are sampled from a Gaussian distribution with mean $g_v$ and $g_\beta$ respectively and standard deviation $\sigma_v$; and with probability $1 - \pi_v$, the genotype of the copies

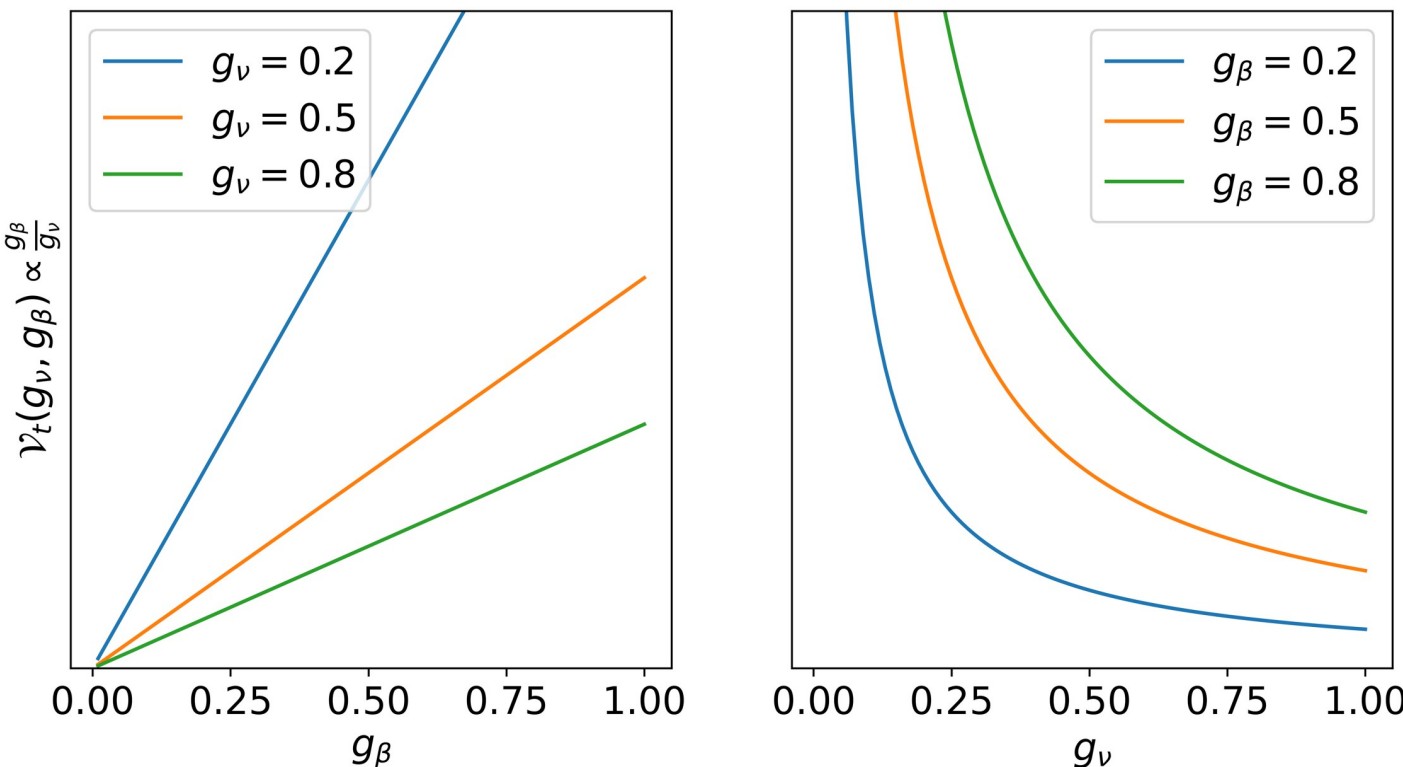

**Fig 2. Trade-off based virulence function $\mathcal{V}_t$.** The left panel shows how $\mathcal{V}_t$ varies with $g_\beta$ for fixed values of $g_\nu$; $\mathcal{V}_t$ is high for high values of $g_\beta$ and low otherwise. The right panel shows how $\mathcal{V}_t$ varies with $g_\nu$ for fixed values of $g_\beta$; $\mathcal{V}_t$ is high for low values of $g_\nu$ and low otherwise.

remains the same as that of the parent. The virus $\nu_i$ cannot infect any more hosts. It is removed from the virus population **V** and the copies of the virus are added to the population.

The behavior of the model is investigated for different parameter values and interaction functions. Sensitivity analyses for virus to host ratios (VHR), limiting resource concentration, and dilution rates are carried out to study the effects of environmental conditions on virus-host dynamics. A range of host and virus mutation probabilities as well as standard deviation for the mutations are considered to see how the arms-race dynamics are influenced by host- and virus-specific cellular constraints. Finally, genotypic and random compatibility/virulence functions are explored in order to analyze the sensitivity of arms-race dynamics to trade-off based virus-host interactions. Additional investigations of the effects of various physiological parameters on virus-host population dynamics are summarized in the supplementary material. In all cases, the effects of different parameters and interaction functions are ascertained from an ensemble average of 100 independent simulations.

## Laboratory experiment

To characterize virus-host infection dynamics in a biological system, as well as to validate the general dynamics emerging from our IBM, we performed an infection experiment with the bacteria *Escherichia coli* E28 (DSM 103246) and a T-4 like virus (DSM 103876, hereby called B28) at five different VHRs (0.01, 0.15, 0.73, 1.90, 3.30). The experiment was conducted in a 96-well flat-bottomed microplate using the 2300 EnSpire Multilabel Plate Reader (PerkinElmer), allowing the entire experiment including multiple replicates to be run simultaneously under one assay. The growth of the host was monitored through automated measurement of

optical density at 600 nm (OD600) every 15 mins. All cultures were grown in the minimal medium M9 containing 2 mM $MgSO_4 \cdot 7H_2O$, 0.1 mM $CaCl_2 \cdot 2H_2O$, 6 mM glucose [27]. All dilutions were also done using M9 medium as diluent.

An outline of the steps involved is as follows: an overnight culture of *E. coli* was prepared, adjusted to OD600 = $\sim 0.2$ using a Cell Density Meter (Fisher Scientific), and further diluted by factor 1:25. A sterile microplate was filled with the host culture, as well as M9 medium as blank solution, final volume of 200 $\mu$L per well. The assay was then run at 37˚C, 150 rpm (linear mode, 3 mm diameter) inside the plate reader. The growth of the host culture was monitored and once the host had entered the exponential growth phase, and its OD600 had increased by 0.04 ($\sim 3.5$ hr), the assay was paused to retrieve the plate. A sample of the host culture was withdrawn and flash-frozen in 20% glycerol for host enumeration at B28 infection timepoint. B28 lysate ($\sim 10^8$ PFU $mL^{-1}$) was then added to the host culture to final volumes of 200 $\mu$L and concentrations of 0.1, 1, 5, 12.5, and 20% v/v. A sample of the lysate was also flash-frozen in 20% glycerol for phage enumeration. The assay was then resumed under the same culture conditions for another $\sim 37$ hr. Host and phage enumeration was later done using a Calibur flow cytometer (Becton Dickinson) following a standard protocol [28], in order to calculate the exact VHRs. The experimental data (`experimental_data.csv`) are available as supporting information.

## Results

In this section, we present host-virus infection dynamics from our growth experiments in the lab (Fig 3), along with results from our IBM (Figs 4–6). The latter are based on an ensemble average of 100 simulations. First we present the effects of environmental conditions on the outcome of the virus-host interaction dynamics in the first 1.5 days after virus infection (Fig 4), then we show the effects of evolution at the cellular level—specifically, mutation probability and mutation variance—(Fig 5), and finally we show the effects of trade-offs in host-virus interactions, considering a longer time-scale of 30 days (Fig 6). All parameter values except those being tested were held constant as shown in Table 1. Parameter values being tested (Figs 4 and 5) are shown in the figure legends. In all figures, host and virus population dynamics are reported in the first and second row, respectively. Note that in each figure, for ease of comparison, we use the same scale for the *y*-axes for hosts and viruses.

Growth experiments with different VHR over the course of 1.5 days reveal that the host population crashes fastest for highest VHR, thereby also reaching a lower maximum population size at the initial population peak. At the same time, the host population recovers fastest for high VHR, resulting in a pronounced second population peak after one day of incubation (yellow, pink and green curves vs black and blue curves in Fig 3). This has the practical consequence that high VHR allows us to observe a full virus-host infection cycle with subsequent recovery of the host within the timeframe of our experiment.

Our IBM captures these dynamics (Figs 4 and 5). Specifically, analogous to the laboratory experiment, high VHR results in earliest crash but also fastest recovery and emergence of potentially resistant hosts in the infected population within the first 36 hours of our in-silico experiments. This manifests as a second peak in host populations reaching between 800 and 1,100 cells $\mu L^{-1}$ roughly 20 hours into the infection cycle (Fig 4a).

Virus numbers increase as they are released from infected hosts and the host population starts collapsing, roughly 10 to 20 hours into the infection cycle. They keep increasing after recovery of the host population, reaching up to 70,000 cells $\mu L^{-1}$. Highest VHR at the start of infection cycle consistently results in higher virus population numbers throughout the simulated time frame (Fig 4d).

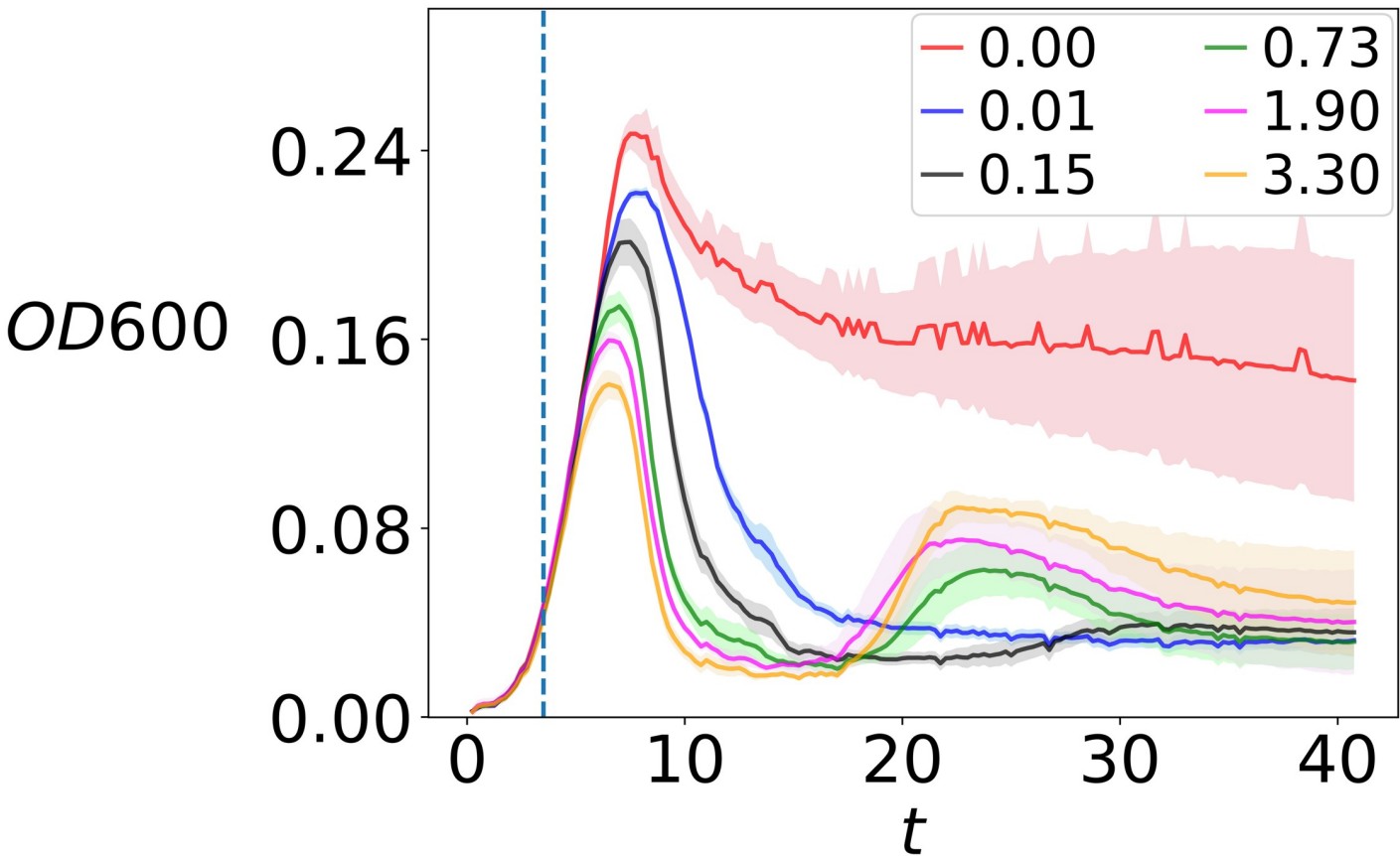

**Fig 3. Infection dynamics of B28 virus and *E. coli*.** Infection experiments performed as plate reader assays for different virus to host ratios (VHR). Optical density at 600 nm (OD600) was monitored to serve as a proxy for the host abundance. The growth curve of *E. coli* without virus infection (control) is shown in red. Plotted curves: mean values ($n = 8$ and $n_{control} = 11$) shown as solid lines and standard deviaitons shown as shading, with the vertical dashed line denoting the time ($t = 3.5$ hr) for viral infection.

Nutrient concentration also influences population dynamics; our model indicates that high availability of nutrients leads to a more pronounced boom and bust scenario in the host population, with hosts reaching higher maximum population numbers in their first peak before they crash (up to 16,000 cells $\mu L^{-1}$) and growing back to high population numbers (around 1,000 cells $\mu L^{-1}$) due to potentially resistant hosts emerging (Fig 4b). Higher limiting nutrient availability is also reflected in highest virus population numbers, which exceeds 80,000 viruses $\mu L^{-1}$ after 36 hours (Fig 4e).

Washout also has a strong effect on population dynamics. A crash in the host population with recovery half-way through the simulated time is most pronounced at low washout rates, reaching numbers down to 300 cells $\mu L^{-1}$ around 18 hours into the infection cycle. Highest washout rates yield no reduction in host population after viral addition, resulting in steady host population numbers of around 1,200 cells $\mu L^{-1}$ from about 12 hours into the infection cycle (Fig 4c). Virus population numbers are highest for low washout rates (Fig 4f). Interestingly, a saturation of the medium with hosts and absence of viruses occurs when the washout value exceeds 0.3 h$^{-1}$ (Fig 4c and 4f).

Besides the effects of environmental parameters (Fig 4), our model shows clear dependence of the virus-host dynamics on the organism-specific evolution of traits, namely mutation probability $\pi$ and standard deviation $\sigma$ of mutations (Fig 5). In particular, increasing the values for

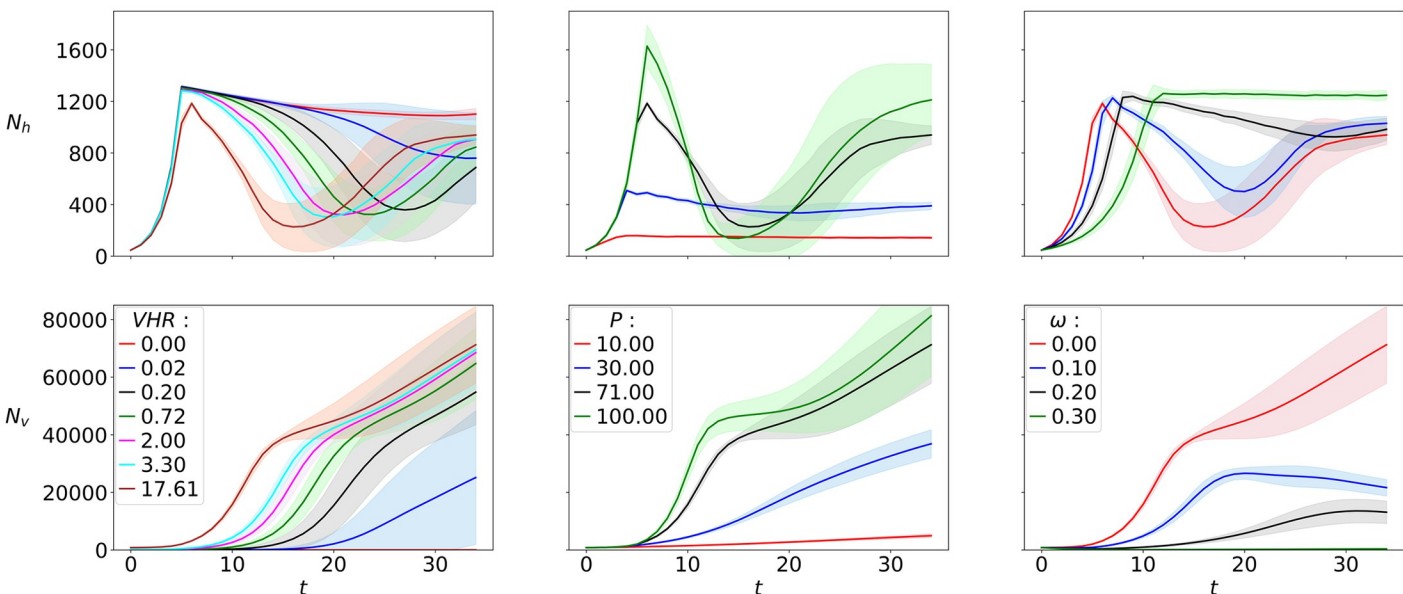

**Fig 4. Model's sensitivity to environmental parameters.** Population dynamics for hosts (a-c) and viruses (d-f) for varying virus to host ratios (VHR), total phosphorous content (P, $\mu$mol-P L$^{-1}$) and washout rate ($\omega$, h$^{-1}$). Plotted curves: ensemble averages from 100 runs lasting $T$ = 36 h, with standard deviations shown as shading.

the host ($\pi_h$ and $\sigma_h$, respectively) leads to a reduced crash and earlier recovery of potentially resistant hosts, with the host population growing back to high densities up to 900 cells $\mu L^{-1}$ within the simulated time (Fig 5a and 5c).

For the lowest tested $\pi_h$, hosts struggle to recover within the simulated time, whereas they collapse entirely for the two lowest tested values $\sigma_h$ (Fig 5a & 5c). Virus population numbers reach high values of up to 80,000 viruses $\mu L^{-1}$ in the simulated time for high $\pi_h$ and $\sigma_h$,

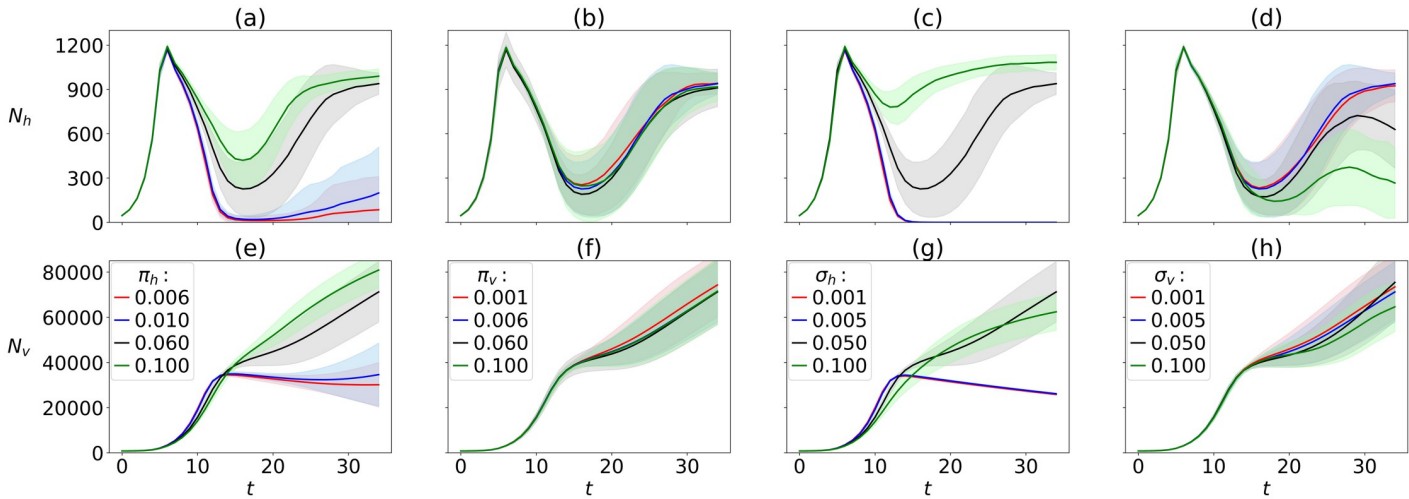

**Fig 5. Model's sensitivity to evolution in organism traits.** Sensitivity of host (a-d) and virus dynamics (e-h) to mutation probability in host traits ($\pi_h$), virus traits ($\pi_v$), standard deviation in mutation for host traits ($\sigma_h$) and standard deviation in mutation for virus traits ($\sigma_v$). Plotted curves: ensemble averages from 100 runs lasting $T$ = 36 h, with standard deviations shown as shading.

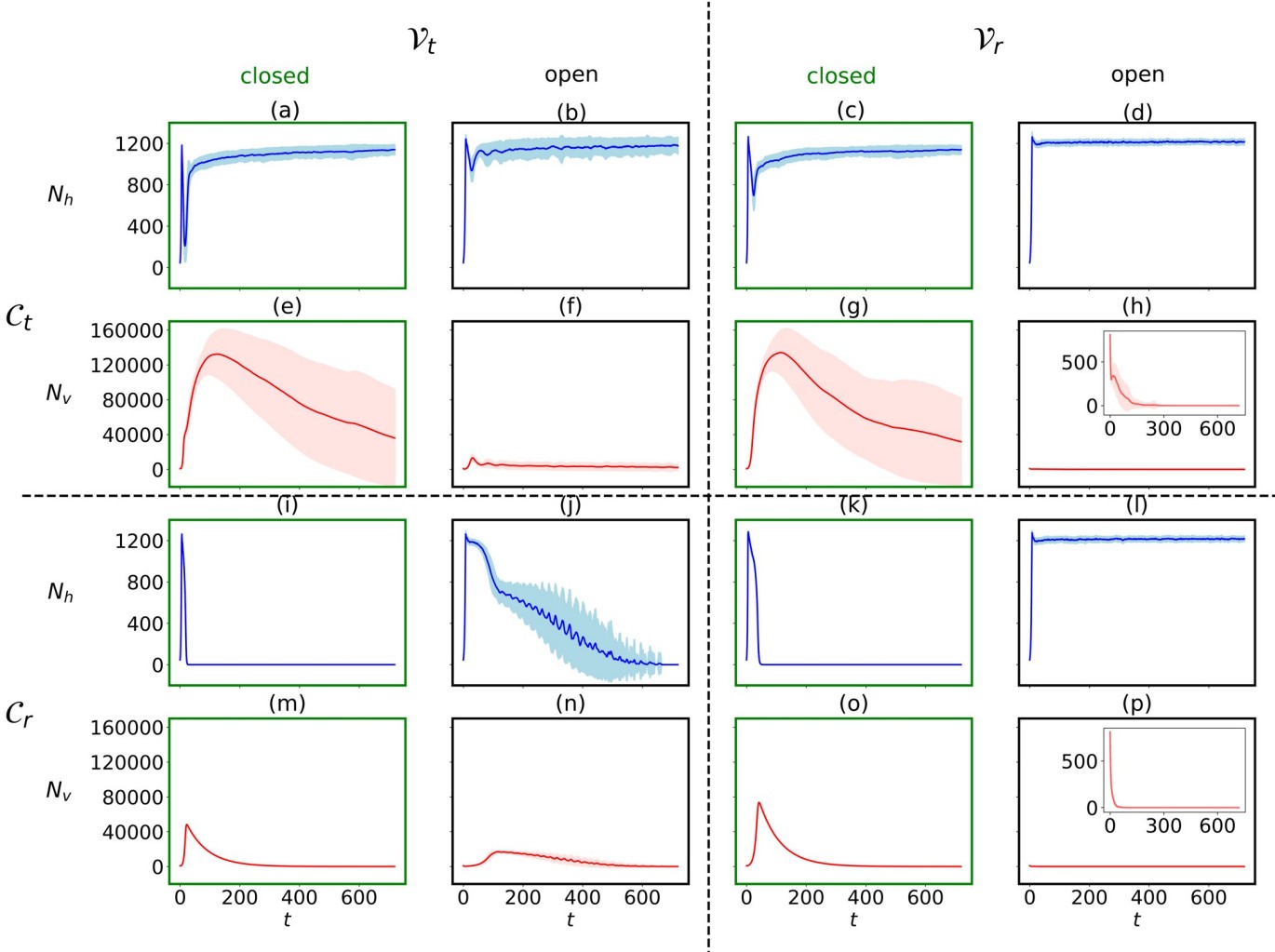

**Fig 6. Model's sensitivity to trade-off-based vs random infection processes.** Host (a-d, i-l) and virus population dynamics (e-h, m-p) shown for closed systems and open systems with washout rate 0.2 h$^{-1}$ for genotypic compatibility and genotypic virulence ($\mathcal{C}_t$, $\mathcal{V}_t$, upper left), genotypic compatibility and random virulence ($\mathcal{C}_t$, $\mathcal{V}_r$, upper right), random compatibility and genotyipc virulence ($\mathcal{C}_r$, $\mathcal{V}_t$, lower left) and random compatibility and random virulence ($\mathcal{C}_r$, $\mathcal{V}_r$, lower right). Ensemble averages of 100 simulations (lines) with standard deviation (shaded areas) lasting for $T = 720$ h are shown. Sub-panels show virus dynamics on adjusted y-axis scales.

increasing readily as the host population recovers from the first crash, whereas low $\pi_h$ and $\sigma_h$ result in stagnant or declining virus population numbers from about 12 hours into infection cycle, never exceeding 40,000 viruses $\mu L^{-1}$ (Fig 5e and 5g).

In contrast to host mutation probabilities, changing the mutation probability for the virus $\pi_v$ over two orders of magnitude does not show any effect on the host-virus dynamics (Fig 5b and 5f). Similarly, increasing the standard deviation for virus mutations $\sigma_v$ does not show a clear effect on virus population dynamics within the simulated time, but the host population recovery after the first crash is dampened with higher $\sigma_v$, reducing the size of the second peak in the host population reaching moderate numbers of 300 cells $\mu L^{-1}$ roughly 28 hours into infection cycle and also allowing for a second crash of the host population within the simulated time of 36 hours (Fig 5d and 5h).

Long-term simulations of 30 days (720 h) reveals marked differences of trade-off based vs random interaction function for virus-host compatibility and virulence (Fig 6). Recall that a trade-off based compatibility function ($C_t$) in our simulations implies that infection success is mediated by a trade-off between the host's nutrient affinity and the virus' adsorpiton coefficient (Fig 1), whereas a trade-off based virulence function ($V_t$) implies that the infection success is dictated by a trade-off between the virus host range and the virus adsorption coefficient (Fig 2).

In all long-term simulations (Fig 6), the system tends towards an equilibrium after the first fluctuations of host crash and typical re-growth thereof that is also visible in the short-term simulations (Figs 4 and 5). The initial crash in host population is most pronounced in closed systems, and hosts as well as viruses die out completely in closed systems under random compatibility modes (Fig 6i, 6m, 6k and 6o).

In simulations with compatibility trade-offs (Fig 6a–6h), hosts reach a high population size at equilibrium after initial drops both in closed and open systems. As in case of random compatibility mode, the first crash of host population is most pronounced in closed systems (Fig 6a and 6c). Viruses reach very high population size at first, independently of whether its virulence is trade-off based or random, but they experience a steady decline over the course of the simulation. Note that the spread in virus population size over the 100 simulation runs is large in closed systems with trade-off based compatibility compared to all other scenarios. In contrast to closed systems, open systems (washout $0.2\,h^{-1}$) suppress the emergence of large viral population all together, keeping them at low numbers after an initial spike when simulations are driven by trade-off based virulence (Fig 6f). For open systems with random virulence, virus population size drop right from the start of the simulation (Fig 6h).

In case of random compatibility (Fig 6i–6p), the host population typically collapses to extinction with the consequence that also viruses eventually die out. The only exception is in open systems where virulence is random. In this case, the viruses disappear early on in the simulation and the hosts persist with high population numbers (Fig 6l). With random compatibility and trade-off based virulence in an open system, host population decreases more slowly and undergoes fluctuations while doing so (Fig 6j). Highest viral population numbers before decline are reached in closed systems. As in open systems with trade-off based compatibility but random virulence (Fig 6h), virus population numbers drop from the beginning of the simulation for open systems and random virulence (Fig 6p).

## Discussion

We have developed a simple individual-based virus-host interaction model to i) study the effects of environmental and organism-specific parameters on population dynamics and ii) evaluate the importance of trade-offs between competitive and defensive host traits on population dynamics and long-term evolutionary change. The model is highly idealized, focusing on specific trait-based mechanisms. Whereas experimental support for trade-offs between competition and defense in diverse biological systems exists [14, 15, 18], other mechanisms such as environmental disturbance and resource specialization also promote diversity. The model is validated with laboratory data from a short-term infection experiment. Some quantitative discrepancy appears between dynamics in the laboratory experiment and our simulations for low VHR, possibly because all viruses in our model are infectious, whereas a fraction of viruses is non-infections in the laboratory. This implies that higher VHR in the laboratory functionally correpsond with lower VHR in our simulations. Regardless of this, the qualitative match between laboratory data and simulation results for host dynamics provides confidence in basic mechanisms underlying virus—host dynamics. Besides, the model produces plausible results

in terms of compatibility trade-offs promoting long-term co-existence of hosts and viruses. However, further validation of the model is needed, in particular for viral population dynamics, which are not as straight forward to measure in laboratory experiments. In the following we discuss details of our findings.

Our model suggests that high VHR results in earliest crash but also fastest recovery of the host population, such that a full infection cycle is complete within the first 36 hours of our simulation experiments. This agrees qualitatively with infection cycles observed in our *E. coli*-B28 experimental model system (Fig 3) where highest VHR also led to fastest regrowth. Interestingly, similar dynamics have been observed previously in marine algal host-virus systems, namely *Emiliania huxleyi* and EhV-99B1, *Pyramimonas orientalis* and PoV-01B, and *Phaeocystis pouchetii* and PpV-01 [29]. Considering selection pressure to be the highest under strong viral control, it is reasonable to assume that re-growth of host populations after the first crash is given by resistant mutants, which readily establish in high VHR conditions. Virus numbers increase simultaneously with the first crash in host populations as they are set free from infected hosts, and keep increasing after recovery of the host population. This suggests that regrowth of the host population is either not exculsively caused by resistant types, but that it is instead diversifying into resistant and susceptible hosts, or that viruses undergo evolutionary change establishing strains that are able to re-infect the originally resistant hosts. The two alternatives not being mutually exclusive, genomic studies will be needed to decipher the cause.

Analogous to microbial blooms under favorable growth conditions, high availability of limiting nutrient gives a boom-and-bust scenario in our simulations with host population reaching very high peaks before crashing to values below those of more stable host population numbers at lower nutrient concentrations (Fig 4b).

Washout rate is also an important parameter for virus-host population dynamics, especially in shorter time-scales, where washout seems to prevent the accumulation of high viral numbers, keeping viral pressure relatively low in open systems (Fig 6, closed vs open system runs). A possible explanation for low viral pressure in open systems could be a combination of washout of viruses and favorable growth conditions for hosts with inflow of fresh medium.

Evolution of organism-specific traits also affects virus-host population dynamics. Our results suggest that variation in host traits may have stronger effects on virus-host population dynamics than variations in virus traits (ie, different host mutation probabilities and standard deviations give more distinct results compared to different viral mutation probabilities and standard deviations, Fig 5). Noticeably, the host population recovery after the first crash is dampened the most with higher standard deviation in mutation for viruses ($\sigma_v$), reducing the size of the second peak in the host population and also allowing for a second crash of the host population within the simulated time (Fig 5d). In other words, high $\sigma_v$ and thus higher flexibility in adjustments of infectious traits as adaptation to acquired host immunity seems to impose a strong selection pressure on hosts, speeding up evolutionary time-scales. Additionally, high $\sigma_v$ should have facilitated the emergence of broader virus host-range, which in nature has been shown to associate with longer interaction periods with hosts [30], potentially making these viruses strong limiting factors for their hosts. Emerging diversity within host and virus specific traits, such as host-range evolution in our model is a subject for a follow-up study. Preliminary results of diversity based on Shannon entropy in distributions of nutrient affinity, virus adsorption coefficient, and memory gene shown in the Supporting Information section indicate increasing diversity over the course of the simulations.

Overall, population dynamics are remarkably different for trade-off based vs random interaction functions. Long-term coexistence of both hosts and viruses is facilitated when the compatibility function is trade-off based (Fig 6a–6h), in which case the viruses persist the longest in closed systems (Fig 6e and 6g). Spatial sturcture has been discussed as mechanism

facilitating long-term co-existence of bacteria and viruses [31], for example by rendering low virulent viruses higher fitness [32]. These findings might explain why eventually, even with compatibility trade-offs, virus populations die out in our well-mixed setting.

The dominance of variation in host traits over virus traits in regulating virus-host population dynamics described in the previous paragraph aligns with the observation that trade-offs involving host traits (ie, compatibility trade-off, which links host traits to virus traits, rather than virulence trade-off that is purely based on viral traits) appeared to have the most pronounced effects effects on the virus-host population dynamics. Indeed, further support for this is provided by our long-term simulations with and without compatibility trade-offs (Fig 6, top half vs bottom half), where trade-off based vs random compatiblity functions result in most distinct long-term dynamics, contrasting trade-off based vs random virulence runs in which difference are only pronounced in open systems (Fig 6, left half vs right half). Interestingly, long-term co-existence also most easily emerge when compatibility trade-offs are expressed. This is true even when viral pressure is high early on in the infection cycle in closed systems. We thus postulate that compatibility trade-offs facilitate virus-host coexistence observed in nature. Interestingly, observations of narrow host range viruses interacting with their hosts at lower total host abundance compared to broad-host range viruses [30] suggest that low host-range interactions are more efficient than those of high host-range viruses, which could be an expression of such a compatibility trade-off playing out in nature. Ranking traits and trade-offs involved in virus-host interaction according to their impacts on evolutionary outcomes is, however, challenging and remains to be studied further.

Our analysis on co-existence of host and viruses supports the hypothesis that trade-offs associated with competitive abilities of hosts and traits influencing the efficiency of viral infection are critical to shaping realistic long-term population dynamics and co-existence in virus-host systems. Competition and defense trade-off being a fundamental phenomenon in nature [33–35], we conjecture that findings from this study might be useful in understanding long-term perseverence of parasites in epidemiological context. Besides, in terms of conditions for high virus population numbers, results might also be useful in pharmaceutical applications, in particular large scale phage production for phage therapy. As a next step, we intend to shed light on host- and virus-population internal trait diversification as a consequence of this trade-off, touching upon the phenomenon of microdiversity in natural microbial communities.

## Supporting information

**S1 Data. Experimental data.** Data on bacteria—phage co-infection experiments rendered in Fig 3 of the main text. The data file inlcudes timeseries of optical density measured at 600 nm, showing number of replicates, mean and standard deviation with 15 minutes interval for different initial virus to host ratios used.
(CSV)

**S1 Pseudocode. Model structures and algorithmic procedures.** The pseudocode describes details of initialization, interactions, budgets and evolutionary dynamics implemented in the model, step by step.
(PDF)

**S1 Fig. Model's sensitivity to physiological parameters.** Host (a-h) and virus dynamics for various resource affinities for hosts ($\alpha$), adsorption coefficients for viruses ($\beta$), maximum growth rates for hosts ($\mu_h$) and burst size for viruses ($\kappa$). Plotted curves: ensemble averages from 100 runs for $T$ = 36 h, with standard deviations shown as shading.
(TIFF)

**S2 Fig. Evolutionary diversification of host and virus genotypes.** Diversity time-series based on Shannon entropy of ensemble averages from 100 runs in a trade-off based compatibiltiy and virulence scenario shown for a) nutrient affinity of host ($g_\alpha$), b) adsorption coefficient of virus ($g_\beta$), and c) memory gene of virus ($g_v$). Small subpanels show abundance distribution for the genotypes at three distinct time points over the course of $T = 720$ h.
(TIFF)

## Author Contributions

**Conceptualization:** Selina Våge.

**Data curation:** Jesslyn Tjendra.

**Formal analysis:** Fateme Pourhasanzade, Swami Iyer.

**Funding acquisition:** Selina Våge.

**Investigation:** Selina Våge.

**Methodology:** Swami Iyer.

**Project administration:** Selina Våge.

**Software:** Fateme Pourhasanzade, Swami Iyer.

**Supervision:** Selina Våge.

**Validation:** Jesslyn Tjendra.

**Visualization:** Fateme Pourhasanzade, Swami Iyer.

**Writing – original draft:** Fateme Pourhasanzade, Swami Iyer, Selina Våge.

**Writing – review & editing:** Fateme Pourhasanzade, Swami Iyer, Jesslyn Tjendra, Lotta Landor, Selina Våge.

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
