## [Decision Letter · Decision Letter 0]

21 Jan 2022

Dear Dr. Våge,

Thank you very much for submitting your manuscript "Individual-based model highlights the importance of trade-offs for virus-host population dynamics and long-term co-existence" for consideration at PLOS Computational Biology.

As with all papers reviewed by the journal, your manuscript was reviewed by members of the editorial board and by several independent reviewers. In light of the reviews (below this email), we would like to invite the resubmission of a significantly-revised version that takes into account the reviewers' comments.

We cannot make any decision about publication until we have seen the revised manuscript and your response to the reviewers' comments. Your revised manuscript is also likely to be sent to reviewers for further evaluation.

Sincerely,

Amber M Smith

Associate Editor

PLOS Computational Biology

Ville Mustonen

Deputy Editor

PLOS Computational Biology

Reviewer's Responses to Questions

**Comments to the Authors:**

Reviewer #1: Pourhasanzade et al. present an interesting manuscript studying how bacteria-phage interactions mediate long-term coexistence of the bacteria and phage. The authors develop a discrete-time individual-based model that includes potential tradeoffs between bacterial growth rates and their ability to defend against infection by phage. They compare modeling results to those of simple infection experiments of E. Coli and a phage at different VHRs on plates.

The population dynamics of E Coli are shown in figure 3. For higher VHRs, the bacteria dynamics are interesting in that the bacteria initially increase then decrease (presumably due to the bacteria-phage interaction) before increasing yet again later on around hour 20. Under certain parameter regimes, the model successfully recapitulates the qualitative nature of these curves.

The individual-based simulations include bacterial growth, death, division, mutation, and virus-host interactions. Generally, I felt that the description of the model could be clarified; perhaps a simple schematic that illustrates the various steps of the model would help a reader understand each element of the model.

I was confused by a couple aspects of the model. I did not see the time step given (is it one hour?). A couple of issues: the parameter pi_h is the probability of mutation of host genotype—surely this must depend on the time step chosen. The parameters delta_h and delta_v are stated as mortality (or decay) rates with units per hour in table 1, but then in the text description (line 76 and line 89) these same parameters are treated as probabilities.

While the model is of course highly idealized, it is still quite complex and contains a large number of parameters (table 1). Many of these parameters are fixed at very specific values given by the study in reference 21. While some sensitivity analyses were performed for a few parameters, it is difficult to know whether model results are specific to this system, or whether results are more general. A broader exploration of parameter space may clarify this.

A major component of the model is the inclusion of bacterial evolution. When a cell divides, its genotype changes with some probability. These genotypes determine the compatibility of the virus and the bacteria through the function C_t. Yet it was difficult to see the connection to data, or to disentangle the effects of evolution from the innate structure of the model (the ecology) in determining population dynamics. Figure 3 shows E Coli population dynamics, but I did not see any sequencing or anything to show whether these dynamics are driven by evolution on the time scale of the experiment. It is difficult to know to what extent evolution is driving these dynamics in the experiments. Moreover, all results in figs 4-6 show population dynamics over time. It is difficult to know to what extent evolution is driving these dynamics in the model. Also, phage population dynamics were not measured, so these cannot be directly compared to the results of the model.

Minor issues:

I would say that Nh and Nv are state variables and not parameters as stated in Table 1.

Typo on line 289 “nutrientt” instead of “nutrient”

I thought the figures could be improved by adding labels to the legends so that the reader would know what is being varied without having to read the caption.

Many of the figures are means of many replicates. It might be nice to shade to show the amount of variation in replicates (standard deviation or similar).

Reviewer #2: See attachment.

Reviewer #3: This paper investigates the hypothesis proposed by others (who are cited in the introduction) that a trade-off between bacterial host competitiveness and host defense against viruses will result in a persistent population of both virus and host. A model is described that takes into account mutation rate, nutrient availability and virus infection efficiency and the results compared to those of an in vivo experiment performed in E. coli with a T-4 like phage. The inclusion of the in vivo experiment is a major strengths of this work in my opinion.

The model parameters are clearly described in tables. The code is not included but is stated to be available upon request. Figures consist of graphs of parameters over time either in the live experiment or the model runs.

Figures are generally clear, but there are ways they or their presentation could be improved.

• It is stated that the model is validated with laboratory data from a short-term infection experiment. However, the in vivo experimental result (Figure 3) and model result (Figure 4a) for host abundance differ considerably at the lowest VHRs and this does not seem to be addressed in the text.

• Methods state that multiple repetitions were done, but no indication of variation between runs is indicated on the graphs and no raw data are provided.

The authors state clearly that their model is “highly idealized, focusing on specific trait-based mechanisms.” However I found myself disagreeing with two assumptions that the investigation is based on, and this lessened my enthusiasm for the work.

Assumption 1: Evolution (genetic diversification) of bacteria requires the long-term co-existence of bacteria and phages.

While it is true that predator-prey relationships are a source of selective pressure that leads to genetic change over time, it is not the only source of selective pressure and therefore not necessary for evolution. Competition for nutrients, as included in the model, are just one of many other selective pressures. The fact that viruses exist and affect population dynamics of their hosts, to me, is not evidence that they are necessary for a healthy ecosystem.

Assumption 2: There is always a trade-off between the (bacterial) host’s growth (competitiveness) and its abilities to defend against the virus. i.e. an increase in virus-resistance host will always result in slower reproductive rate.

If taken to the mathematical extreme, doesn’t this assumption imply that a host with 100% virus resistance should be completely non-competitive with regard to growth? But absolute resistance to phage is known to exist in viable bacterial strains, which means in a single virus-host pair scenario, the virus would go extinct. Doesn’t this disprove the single host-virus pair model?

minor typo: virues is missing an "r" in the abstract

**Have the authors made all data and (if applicable) computational code underlying the findings in their manuscript fully available?**

Reviewer #1: **No: **I did not see that the code for the simulations and the raw data from the laboratory experiment are made available.

Reviewer #2: **No: **Code is available upon request, but could probably be made available more immediately via SI or online repository.

Reviewer #3: **No: **The data points behind the plotted means and variance measures were not provided and there was no indication of a publicly accessible repository of the information or code.

PLOS authors have the option to publish the peer review history of their article (what does this mean?). If published, this will include your full peer review and any attached files.

Reviewer #1: No

Reviewer #2: No

Reviewer #3: No
---

## [Decision Letter · Decision Letter 1]

25 Apr 2022

Dear Dr. Våge,

Thank you very much for submitting your manuscript "Individual-based model highlights the importance of trade-offs for virus-host population dynamics and long-term co-existence" for consideration at PLOS Computational Biology. As with all papers reviewed by the journal, your manuscript was reviewed by members of the editorial board and by several independent reviewers. The reviewers appreciated the attention to an important topic. Based on the reviews, we are likely to accept this manuscript for publication, providing that you modify the manuscript according to the review recommendations.

Sincerely,

Amber M Smith

Associate Editor

PLOS Computational Biology

Ville Mustonen

Deputy Editor

PLOS Computational Biology

[LINK]

Reviewer's Responses to Questions

**Comments to the Authors:**

Reviewer #1: I thank the authors for their detailed responses. Most issues were adequately addressed. The authors now include supplementary pseudocode, a GitHub link to their code, and a more detailed model description, all of which make it far easier for a reader to follow their methodology. The figures are also greatly improved. One minor remains issue with the figures: Fig 3-5 are missing an explanation as to the meaning of the shading.

A key limitation that still exists is that phage population dynamics are not measured so that they cannot be directly compared to model results. In lines 284-287 of the discussion, this is touted as an advantage of the model. But I would argue that it is only an advantage if the viral population dynamics are independently verified with a rigorous comparison with data. Before such verification, the phage population dynamics seem speculative. In my view, such measurements are not needed for publication of this manuscript as the authors already state the challenges associated with obtaining such measurements. However, I do think that this limitation should be acknowledged.

Additional minor issue: Figure S2 does not appear to be referred to in the main text.

Reviewer #2: The authors seem to have addressed the majority of reviewer questions and concerns. While we might want a few more answers now, this paper seems to lay the groundwork for followup work.

I am satisfied with the revision and especially updates to the model description, tables/notation, and figures.

** I missed this the first time, but on line 145 a dot/bullet symbol is used in some chemistry notation. Not sure if this is standard.

**Have the authors made all data and (if applicable) computational code underlying the findings in their manuscript fully available?**

Reviewer #1: Yes

Reviewer #2: Yes

PLOS authors have the option to publish the peer review history of their article (what does this mean?). If published, this will include your full peer review and any attached files.

Reviewer #1: No

Reviewer #2: No

Figure Files:

Data Requirements:

Reproducibility:

References:

---

## [Editor Report · Decision Letter 2]

17 May 2022

Dear Dr. Våge,

We are pleased to inform you that your manuscript 'Individual-based model highlights the importance of trade-offs for virus-host population dynamics and long-term co-existence' has been provisionally accepted for publication in PLOS Computational Biology.

Best regards,

Amber M Smith

Associate Editor

PLOS Computational Biology

Ville Mustonen

Deputy Editor

PLOS Computational Biology

---

## [Editor Report · Acceptance letter]

2 Jun 2022

PCOMPBIOL-D-21-02137R2 

Individual-based model highlights the importance of trade-offs for virus-host population dynamics and long-term co-existence

Dear Dr Våge,

I am pleased to inform you that your manuscript has been formally accepted for publication in PLOS Computational Biology. Your manuscript is now with our production department and you will be notified of the publication date in due course.

With kind regards,

Zsofia Freund
